# Overexpression of *ApHIPP26* from the Hyperaccumulator *Arabis paniculata* Confers Enhanced Cadmium Tolerance and Accumulation to *Arabidopsis thaliana*

**DOI:** 10.3390/ijms242015052

**Published:** 2023-10-10

**Authors:** Lizhou Zhou, Lvlan Ye, Biao Pang, Yunyan Hou, Junxing Yu, Xuye Du, Lei Gu, Hongcheng Wang, Bin Zhu

**Affiliations:** School of Life Sciences, Guizhou Normal University, Guiyang 550025, China; 21010100413@gznu.edu.cn (L.Z.); 222100100456@gznu.edu.cn (L.Y.); 21010100406@gznu.edu.cn (B.P.); 21010100376@gznu.edu.cn (Y.H.); yujunxing@gznu.edu.cn (J.Y.); duxuye@gznu.edu.cn (X.D.); 201808009@gznu.edu.cn (L.G.)

**Keywords:** Cadmium stress, *Arabis paniculata*, *ApHIPP26*, transcriptome, phytohormone pathway

## Abstract

Cadmium (Cd) is a toxic heavy metal that seriously affects metabolism after accumulation in plants, and it also causes adverse effects on humans through the food chain. The HIPP gene family has been shown to be highly tolerant to Cd stress due to its special domain and molecular structure. This study described the Cd-induced gene *ApHIPP26* from the hyperaccumulator *Arabis paniculata*. Its subcellular localization showed that ApHIPP26 was located in the nucleus. Transgenic *Arabidopsis* overexpressing *ApHIPP26* exhibited a significant increase in main root length and fresh weight under Cd stress. Compared with wild-type lines, Cd accumulated much more in transgenic *Arabidopsis* both aboveground and underground. Under Cd stress, the expression of genes related to the absorption and transport of heavy metals underwent different changes in parallel, which were involved in the accumulation and distribution of Cd in plants, such as *AtNRAMP6* and *AtNRAMP3*. Under Cd stress, the activities of antioxidant enzymes (superoxide dismutase, peroxidase, catalase, and ascorbate peroxidase) in the transgenic lines were higher than those in the wild type. The physiological and biochemical indices showed that the proline and chlorophyll contents in the transgenic lines increased significantly after Cd treatment, while the malondialdehyde (MDA) content decreased. In addition, the gene expression profile analysis showed that *ApHIPP26* improved the tolerance of *Arabidopsis* to Cd by regulating the changes of related genes in plant hormone signal transduction pathway. In conclusion, *ApHIPP26* plays an important role in cadmium tolerance by alleviating oxidative stress and regulating plant hormones, which provides a basis for understanding the molecular mechanism of cadmium tolerance in plants and provides new insights for phytoremediation in Cd-contaminated areas.

## 1. Introduction

Drought, extreme temperature, high salinity and heavy metals are the most serious abiotic stresses affecting plant growth [1]. Heavy metals (HMs) are one of the most critical abiotic factors affecting soil and water and have become a serious worldwide problem [2]. They are considered to be the most serious pollutant in ecotoxicology. Although HMs exist in nature, due to the rapid development of industry their content is increasing, resulting in great adverse effects [3,4]. Cadmium (Cd) is a nonessential element for plants and has become one of the most toxic pollutants in water and soil worldwide [5]. It not only causes soil dysfunction and water quality decline but also damages the physiological development of plants to varying degrees, such as reducing chlorophyll, inhibiting nutrient absorption, and causing cell death, thus affecting the growth and metabolism of plants [6,7,8,9]. Cd is absorbed by plants, enriched and transferred into the food chain, endangering human life and health [10,11,12]. Considering the threat posed by Cd to crop production and quality as well as human and animal health, reducing its accumulation is essential. Therefore, it has been an urgent and high-priority issue to elucidate the molecular mechanism of Cd tolerance in plants and to breed or engineer crops with higher tolerance but less accumulation of Cd.

Because Cd has a similar structure to other essential elements (Ca, Zn, and Fe), the transporter transports these essential elements while transporting Cd into the plant. As a heavy metal, a large amount of Cd accumulation will have a toxic effect on the plant. [12,13]. Heavy metals enter the roots of plants from the soil through complex pathways and are then transported to the aboveground parts. This process involves the participation of some heavy metal-related transporters, including iron-regulated transporter (IRT), zinc-regulated transporter/IRT-like protein (ZIP), natural resistance-associated macrophage protein (NRAMP) and heavy metal ATPase (HMA) [14,15].

HIPPs (heavy metal-associated isoprenylated plant proteins) are unique metallochaperone proteins in vascular plants and are mainly involved in the reaction of heavy metal homeostasis, detoxification mechanisms, heavy metal tolerance (especially Cd tolerance), cold resistance, drought resistance and other mechanisms [16]. These proteins generally contain 113 to 584 amino acid residues and have one or two heavy metal-associated domains (HMAs), of which the core motif is M/LXCXXC (M for methionine, L for leucine, X for amino acids, C for cysteine), with the ability to bind to Cu, Ni, and Zn [17]. A typical structural feature of HIPPs is that the C-terminus has a CaaX structure (C represents cysteine; a represents aliphatic amino acids; X represents methionine, glutamine, cysteine, serine, or alanine), which is crucial for biological functions with protein–membrane and protein-protein interactions [18]. The function of HIPPs is apparent but only characterized in a handful of plants. Whole-genome analysis showed 56, 54 and 13 HIPP genes in *Camellia sinensis* (L.) Kuntze [19], *Oryza sativa* L. [20], and *Fagopyrum tataricum* (L.) Gaertn. [21], respectively. *CsHIPP24* overexpressed in Cd-sensitive yeast (ycf1) displayed higher performance than *CsHIPP22*/*36* in Cd resistance. Overexpression of the *CdI19* gene enhances the blocking effect of the endoplasmic reticulum and reduces the damage of cadmium to cells in *Arabidopsis* [22]. The overexpression of *OsHIPP56* exhibited a remarkable detoxification effect with an up to 230% reduction in Cd deposition in rice roots compared with WT [23]. The expression level of *OsHIPP21* was increased by Cd induction, as was the case for *OsHIPP28* and *OsHIPP41* [24]. The transformed yeast mutants expressing *OsHIPP16*/*34*/*60* accumulated more heavy metals (treated with Mn, Cu, Cd and Zn, respectively) without growth inhibition, which was validated by rice transformation of *OsHIPP42* [18]. *AtHIPP44* can interact with the transcription factor *AtMYB49* and increase its expression, thereby reducing Cd accumulation [25]. Although there are many reviews on HIPPs, the functional verification of *HIPP26* is lacking.

It has been reported that *Arabis paniculata* Franch is a new type of heavy metal hyperaccumulator that has a strong enrichment effect on Zn, Cd and Pb [26]. In our previous study, we sequenced the transcriptome of *A. paniculata* under Cd stress and found that the *HIPP26* gene was significantly upregulated [27]. However, the role of this gene in the physiological response and molecular mechanism to Cd stress is unclear. Therefore, the *ApHIPP26* gene was cloned and overexpressed in *Arabidopsis thaliana* (L.) Heynh to verify the molecular mechanism under Cd stress. Overexpression of *ApHIPP26* enhanced Cd tolerance and accumulation by regulating plant hormone signal transduction in *A. thaliana*.

## 2. Results

### 2.1. ApHIPP26 Conserved Motif and Phylogenetic Analysis

The ApHIPP26 protein was conserved among the 10 species of plant. As shown in Figure 1a, the ApHIPP26 protein is evolutionarily close to that of *C. sativa*. Motif analysis showed that the motif of ApHIPP26 is overwhelmingly identical to that of ApHIPP26 proteins in other plant species (Figure 1b).

### 2.2. ApHIPP26 Is Localized in the Nucleus

To determine the subcellular localization of ApHIPP26, we injected ApHIPP26-GFP and GFP expression vectors into tobacco epidermal cells for transient expression. The GFP signal and DAPI signal completely overlapped, while the negative control signal was distributed in the cell membrane, cytoplasm and nucleus (Figure 2). These results indicate that ApHIPP26 is localized in the nucleus.

### 2.3. Cd Induced ApHIPP26 Expression

RT-qPCR results showed that there was no difference in the expression level of ApHIPP26 in roots and shoots of *A. paniculata* under normal conditions. After exposure to Cd, the mRNA levels of roots and shoots increased significantly, especially in the latter part (Figure 3a). The results show that the *ApHIPP26* gene is a key gene of *A. paniculata* in response to Cd stress, and its expression is upregulated when Cd stress occurs.

### 2.4. ApHIPP26 Overexpression Enhanced Cd Tolerance and Accumulation in Transgenic Arabidopsis

We obtained nine transgenic *Arabidopsis* lines by inflorescence infection, and fluorescence quantitative RT-qPCR was performed to detect *ApHIPP26* expression in these nine lines. Three lines (OE2, OE5 and OE9) with high expression levels were options for further operation (Figure 3b). In a normal medium, there was no significant difference in the phenotype (root length, fresh weight, germination rate) of wild-type (WT) and transgenic lines. However, in the medium with 100 μM Cd, the root length of the wild type (WT) was significantly shorter than that of the three transgenic lines with the same performance in both fresh weight and germination rate, of which the wild type was significantly lower (Figure 4a,b and Figure 5a–c). The results of 14-day soil incubation showed no difference in the phenotypes of WT and transgenic lines under normal conditions, but the leaves of WT showed much more severe yellowness than those of transgenic plants treated with 2.5 mM Cd (Figure 4c), which indicated that the overexpression of *ApHIPP26* enhanced Cd tolerance in transgenic *Arabidopsis*.

### 2.5. Overexpression of ApHIPP26 Enhanced the Antioxidant Capacity and Photosynthesis of Transgenic Arabidopsis

We detected the contents of MDA and proline (Pro), which can reflect the degree of plant injury under stress. In this study, the MDA content of WT plants was significantly higher than that of transgenic lines stressed by Cd (Figure 6a), and the content of proline in WT plants showed a downwards trend compared to transgenic plants (Figure 6b), indicating that the cell membrane of WT plants had suffered more serious damage. Meanwhile, the antioxidant system in the plant functioned, eliminating some ROS to mitigate the effects of stress. The POD, APX, CAT, and SOD enzymes in OE plants exhibited significantly higher activity than those in WT plants (Figure 6c–f). Moreover, the chlorophyll content was rescued in transgenic plants and was significantly higher than that in WT plants under Cd stress (Figure 6g). These results implied that overexpression of *ApHIPP26* in *Arabidopsis* enhanced Cd tolerance by improving the antioxidant capacity and rescuing photosynthesis.

### 2.6. ApHIPP26 Increased Cd Uptake by Regulating the Expression of Heavy Metal Transporters

After Cd exposure, the Cd content of shoots and roots in the OE5 and WT lines indicated that overexpression of *ApHIPP26* increased Cd uptake both aboveground and underground (Figure 7i).

Plants have developed a regulatory system to maintain the balance of metal ion concentrations, among which heavy metal transporters are an important component and include the yellow stripe-like transporter protein family (YSL), ZRT/IRT-like protein family (ZIP), natural resistance-associated macrophage protein (NRAMP), etc. We selected eight genes encoding related proteins to verify whether overexpression of *ApHIPP26* regulates their expression. In the absence of cadmium treatment, overexpressed *ApHIPP26* elevated the expression level of *AtNRAMP6* and *AtNAS3* and decreased that of *AtHMA2*, with no difference in others. Under Cd treatment, *AtNRAMP6* and *AtNAS3* were expressed at significantly higher levels, and *AtNRAMP3*, *AtYLS2*, *AtIRT3*, *AtZIP4*, and *AtCAX3* were downregulated.

### 2.7. ApHIPP26 Enhanced the Synthesis of Plant Hormones in the Aerial Parts of Transgenic A. thaliana

To further understand the regulatory network after the *ApHIPP26* gene was introduced into *A. thaliana*, WT and OE5 were treated with 0 mM and 2.5 mM, and the DEGs of their aboveground parts were identified. There were 223 DEGs in transgenic lines in the comparison with Cd treatment and without (CKOE5/CdOE5), and 4652 EDGs in WT lines treated with and without Cd (CKWT/CdWT). In addition, a total of 2928 DEGs were identified in Cd-treated WT and transgenic lines (CdOE5/CdWT), while 1217 DEGs were identified in the control group without Cd treatment (CKOE5/CKWT). The results showed that the *ApHIPP26* gene altered the transcription profile of the Cd stress response after its transfer into *Arabidopsis* (Figure 8a).

Subsequently, we performed GO analysis to classify the biological functions of the DEGs identified in CdOE5 vs CdWT. We found that many of these DEGs were enriched in the lignin biosynthetic process, cell wall macromolecule biosynthetic process, and plant-type cell wall modification (Figure 8b). These biological processes are closely related to plant cell wall synthesis, and the DEGs were all downregulated. In the KEGG pathway, DEGs were mainly enriched in the starch and sucrose metabolism, plant hormone signal transduction, and cysteine and methionine metabolism pathways (Figure 8c). Noticeably, we found that the expression levels of DEGs in the plant hormone signal transduction pathway changed significantly (Appendix A), so we analysed them (Figure 8d). Twenty, ten and three DEGs were detected in auxin, abscisic acid (ABA) and ethylene (ETH) signal transduction, respectively. *AtIAA27* and *AtARF11* were upregulated, and *AtHAB1*, *AtABI2* and *AtERF1* were downregulated (Figure 9a–h).

## 3. Discussion

Cd can pollute soil, water, air dust and food in various ways, which causes Cd poisoning in people and animals, mainly by feeding or inhalation, and cadmium compounds are the most severe. Cd poisoning mainly manifests as kidney damage, osteoporosis, anemia, nervous system damage, etc., and severe cases can lead to kidney failure and death. Human chromosome aberrations caused by chronic poisoning seriously affect offspring growth and development [28]. To date, many technologies for the remediation of heavy metal-contaminated soil are mainly based on physical and chemical principles [29,30,31]. However, some limitations remain, such as high cost and low efficiency when pollutants are present at low concentrations [31,32,33]. Therefore, it is necessary to develop more economical and efficient technologies. Phytoremediation is plant-based remediation that can absorb metallic elements from soil and water, accumulate them in plants and purify the environment [34]. Recently, the study of hyperaccumulators has shed new light on these issues of heavy metal pollution and has become one of the research highlights.

According to professional evaluation and analysis, many star plants applied in phytoremediation have been found, and molecular mechanisms and useful candidate genes for the absorption and accumulation of heavy metals have been explored in these plants. In the Populus×canescens line, the *CTP3* gene from *Sedum plumbizincicola* X. H. Guo was overexpressed and increased cadmium tolerance and accumulation, altering the distribution of Cd in leaf tissues [35]. Moreover, RNAi and overexpression techniques proved that *SpHMA3* positively regulated Cd accumulation by compartmentalization in *Sedum plumbizincicola* [36]. Overexpression of the root-specific expression gene *PCR2* in *Sedum alfredii* Hance enhanced the Cd exocytosis of roots, which was confirmed in *A. thaliana* [37]. Ectopic expression of *ZNT1* cloned from *Noccaea caerulescens* (J.Presl & C.Presl) F.K.Mey. revealed that *NcZNT1* contributed greatly to root-shoot transport of metals such as nickel, Cd, zinc, iron (Fe), and manganese (Mn) in *A. thaliana* [38]. The upwards transport of Fe and Mn rather than Cu or Zn was increased in transgenic *A. thaliana* overexpressing *SnYSL3* (*Solanum nigrum* L.), but when stressed by Cd the transport efficiency of Mn no longer changed [39]. *MsYSL1*, one upregulated gene in *Miscanthus sacchariflorus* treated with Cd, manages the transport of Cd, Fe, and Mn and is involved in the process of Cd detoxification by inducing the expression of genes related to nicotianamine synthesis and HM exclusion [40]. Previously, we executed one transcriptome study and found that the *ApHIPP26* gene in the hyperaccumulator *A. paniculata* is induced by Cd stress [41]. In this study, we ectopically expressed *ApHIPP26* in *Arabidopsis* to further explore its function in the response to HM stress.

Reactive oxygen species (ROS), secondary metabolites produced in aerobic metabolism, are recognized as key signal transduction molecules that are deeply involved in programmed cell death, growth and development, hormone signaling, and responses to various environmental stresses [42]. However, excessive ROS mainly attack biological macromolecules (membrane lipids, proteins and nucleic acids) within cells, producing cytotoxic effects and even triggering rapid cell death [43]. The generation and digestion of ROS in cells must remain in dynamic equilibrium, which is the basis of maintaining a relatively stable redox state for survival. As the final product of membrane lipid peroxidation, MDA content increases with increasing lipid membrane oxidation damage, and Pro can regulate permeability and prevent peroxidation, reflecting the antioxidant capacity of plants [44]. SOD, POD, and CAT are considered important scavengers and components of antioxidant defense in plants [45]. In this study, transgenic plants overexpressing *ApHIPP26* exhibited stronger antioxidant capacity under Cd stress, with excellent antioxidant oxidase activity (Figure 6), and the content of MDA significantly decreased, while the WT plants suffered more seriously from peroxide. Of course, overexpression of *ApHIPP26* also increased root growth and biomass, which is important for plant growth (Figure 5).

The most important feature of hyperaccumulator plants is their ability to absorb heavy metals and transfer them to aerial parts [46]. In this study, overexpression of *ApHIPP26* significantly increased the uptake and translocation of Cd in *Arabidopsis* (Figure 7b). Therefore, we detected some genes known to be involved in the above processes and found that the expression of *AtNRAMP6* and *AtNAS3* was significantly upregulated, while that of *AtNRAMP3*, *AtYLS2*, *AtIRT3*, *AtZIP4*, and *AtCAX3* was downregulated. *SaNRAMP6* from *Sedum alfredii*, when transferred to yeast and *Arabidopsis*, enhances their sensitivity to Cd by increasing Cd transportation and accumulation [47]. In tobacco, knocking out the *NtNRAMP3* gene makes it easier for Cd to accumulate in vacuoles, altering the distribution of Cd in subcellular cells of leaves [48]. Due to the overexpression of *ApHIPP26*, the expression of genes related to heavy metal absorption and transport undergoes different changes in parallel, jointly participating in the accumulation and distribution of Cd in plants. This also confirms that overexpression of *ApHIPP26* enhances cadmium accumulation.

In hyperaccumulators, the main transport site of metals is the xylem, in which heavy metals form complexes with various organic acids or amino acids and are translocated upwards [49]. In this situation, the metal ions deposited in the cell wall are not easily transported [50]. After entering leaf tissues, the metal mainly crossed the cell wall and became compartmentalized in the vacuole [2]. Our GO analysis revealed that DEGs are mainly classified into cell wall synthesis and are all downregulated, which makes it easier for heavy metals to enter rather than bind to the cell wall. Additionally, our KEGG analysis revealed that DEGs were enriched in cysteine and methionine metabolism pathways (Figure 8c). As reported, sulfur-containing methionine and cysteine achieve transport and detoxification of heavy metals through phytochelatins and metallothioneins, and these chelates and complexes can reduce the toxicity of heavy metals and ROS by the oxidation reaction of their thiol group (-SH) [51,52]. Exogenous cysteine can improve the tolerance of plants (maize, *Arabidopsis*) to heavy metals such as Cd, chromium and mercury by increasing the activity of antioxidant enzymes and regulating the expression of *GS1* and *MT3* genes, which encode glutathione and metallothioneins [53]. Precursors involved in cysteine and methionine biosynthesis are also closely related to abiotic stress in plants. The CSase gene, belonging to the cysteine synthase gene family and involved in cysteine synthesis, was overexpressed in Medicago sativa and significantly increased the content of cysteine and glutathione, which contributed to Cd resistance in alfalfa [54]. *SAMS1*, one gene encoding S-adenosyl-methionine synthetase, enhanced the salt tolerance of both tobacco and *Arabidopsis* by polyamine synthesis [55], which was verified by the protein interaction of *CaM4* with *SAMS1* governing ion homeostasis and ethylene synthesis [56].

Phytohormones, as endogenous substances, play significant roles in modulating plant germination, growth, flowering, fruit and other processes, especially at very low concentrations, and their functions are complex and sophisticated [57,58]. The phytohormone strategy used to solve HM poisoning in plants has attracted strong attention [59], including foliar application of exogenous phytohormones and endogenous regulation of auxin, abscisic acid (ABA), ethylene (ETH) and others [60,61,62]. Auxins help plants respond to heavy metal toxicity by regulating their biosynthesis, degradation, signal transduction and transport and play an important role in root development under normal and stress conditions [63]. Auxin plays an important role in root development under normal and stress conditions. In tomato, *SI-IAA27* was identified as a pivotal gene that passes real-time information between auxin, ethylene biosynthesis and strigolactone biosynthesis through transcription regulation modules such as *SI-IAA27*-SI-ERF. B3 and *SI-IAA27*-SI-NSP1 [64]. Moreover, overexpressing *ApHIPP26* inhibited the activity of the ethylene pathway, in which three genes and the key gene in ethylene synthesis, *ACS2*, were all downregulated, which is consistent with the results showing that inhibiting ethylene synthesis can enhance cadmium tolerance in plants [65,66]. In our study, *AtIAA27* (AT4G29080) in auxin and *AtERF1* (AT3G23240) in the ethylene pathway were differentially expressed. The ABA signaling pathway consists of several core components (e.g., Snfl-related protein kinase 2, SnRK2s; 2C-type protein phosphatase, PP2Cs; ABA-response element binding factors and ABF), which play an important role in the response to abiotic stress [67]. The transcription factor *MYB49* promotes the accumulation of Cd by binding the promoter of *HIPP22* to *HIPP44*, while the accumulation of Cd in *Arabidopsis* leads to an increase in endogenous ABA content, which induces *ABI5* expression and inhibits *MYB49* binding to downstream regulatory genes to achieve negative feedback regulation [25]. *ABI5* interacts with *HAB1*/*ABI2* by dephosphorylation [68]. In this work, DEGs (*HAB1*/*ABI2*) related to the negative regulator PP2C were downregulated, which further activated downstream ABF-binding factors, thereby enhancing ABA signal transduction, which suggested that ABA signal transduction was activated by overexpression of *ApHIPP26* and enhanced Cd accumulation. However, this conjecture contradicts some research findings that high ABA levels enable Cd deposition in apoplastic obstacles and inhibit the transport of Cd via apoplastic pathways [69,70]. We believe that whether ABA promotes metal accumulation depends on the metal concentration and plant species, and the mechanism is still complex [71].

## 4. Materials and Methods

### 4.1. Plant Culture and Treatments

The materials used in this experiment were *A. thaliana*, *A. paniculata* and *Nicotiana tabacum* Linn. (Huaxi District, Guiyang, China, E106.67°, N26.41°). Tobacco seeds were directly sown in a square box filled with nutrient soil and the seedlings were transplanted after two weeks. *Arabidopsis* seeds were washed with alcohol and then washed with sterile water 3 times, then were vernalized at 4 °C for 3 days, germinated vertically on a Murashige and Skoog (MS) medium for 7 days. After three weeks of growth in the nutrient soil, six pots of wild-type *Arabidopsis* and three transgenic lines (four plants per pot) were selected and divided into two groups. The Hoagland medium with or without 2.5 mmol/L CdCl_2_ was used to irrigate for 14 days. The seeds of *Arabidopsis* were washed in the same way and root length and germination rate were measured on media with or without 100 μm CdCl_2_ and 150 μm CdCl_2_ on a 1/2 Murashige and Skoog (1/2MS) medium. Three biological replicates were set for each group. After three weeks in Hoagland solution, *A. paniculata* seedlings were treated with 0.25 mM CdCl_2_ for seven days. The artificial climate incubator strictly implements the light and humidity control system with specific indicators referring to a published article [66].

### 4.2. Phylogenetic Analysis

Using the protein sequence translated from the coding sequence of ApHIPP26 as a query condition, the homologous ApHIPP26 protein of *A. thaliana*, *Eutrema salsugineum* (Pall.) Al-Shehbaz, *Brassica rapa* L., *Capsella rubella* Reut. ex Boiss., *Raphanus sativus* L., *Camelina sativa* (L.) Crantz, *Brassica oleracea* L., *Tarenaya hassleriana* (Chodat) H.E. Moore, *Salvia splendens* Sellow ex Roem., *Phtheirospermum japonicum* (Thunb.) Kanitz, and *Solanum lycopersicum* L. MEGA7.0 was used to analyse the phylogenetic relationships of HIPP26 proteins among these 11 species. DNAMAN was used to analyse the multiple sequence alignment of HIPP26 protein among these 11 species.

### 4.3. Subcellular Localization

Primers were designed according to the CDs sequence of the *HIPP26* gene and the pROKII vector with restriction sites. The pROKII-ApHIPP26-GFP vector was constructed by homologous recombination and transformed into *Agrobacterium* GV3101 by the freeze-thaw method. The prepared bacterial liquid was injected into the tobacco leaves, and the tobacco seedlings were put into an artificial climate incubator for dark treatment and cultured for two days. The blade was cut into 1 cm^2^ pieces to make temporary slides, which were observed and photographed with a laser confocal microscope (Zeiss, LSM510 Meta, Carl Zeiss). The pROKII empty vector was used as the control.

### 4.4. RNA Isolation and qRT–PCR

Total RNA extraction was performed with an RN38-EASYspin Plus Kit (Aidlab, Hangzhou, China) and RT Master Mix for qPCR II Kit (MedChemExpress, Shanghai, China) for reverse transcription. The BIOER FQD-48A system (BIOER, Hangzhou, China) was utilized for RT-qPCR. The gene sequences of primers used are shown in the attached table, and the reference gene is Actin 1. The 2^−ΔΔCT^ method, which was reported previously [72], was employed to estimate gene expression.

### 4.5. Construction of the Expression Vector and Genetic Transformation of A. thaliana

The construction of the pB121-HIPP26 vector was based on a published method [28]. A. thaliana growing to bloom was used for genetic transformation mediated by *Agrobacterium* GV3101 [29]. Screening of positive plants was carried out by 1/2 MS medium containing 50 mg/L kanamycin, and T3 seedlings were obtained for subsequent study.

### 4.6. Phenotypic Analysis and Determination of Physiological Indices

The antioxidant enzymes in aboveground parts were determined, including superoxide dismutase (SOD), peroxidase (POD), catalase (CAT), ascorbate peroxidase (APX), and the levels of malondialdehyde (MDA), proline (Pro), and chlorophyll. All substances were quantified using a detection kit (Solarbio, Beijing, China) that contained the details of the experimental procedure.

### 4.7. Determination of Cd Content

Cd accumulation in the shoots and roots of the transgenic lines (OE2, OE5 and OE9) and wild type (WT) was determined. It should be noted that the roots of the plant should be sufficiently cleaned to ensure that Cd in solution was removed. The plant samples were dried and ground into powder, with specific operational details referring to published papers [73], and then concentration measurements were performed using ICP-MS [74].

### 4.8. RNA-Seq Analysis

We selected aboveground tissues of transgenic plants and WT plants for transcriptome sequencing. The two groups were treated with different concentrations of Cd (0 mM CdCl_2_, 2.5 mM CdCl_2_). A NanoDrop 2000 spectrophotometer, which is a professional platform to quantify the samples, was initially used to quantify the RNA concentration. Then, an Agilent 2100/4200 was used to interpret the RNA sample quality and accurately quantify the concentration. When the construction of the library was finished, Qubit 3.0 software was available and used for initial quantification. The Illumina NovaSeq 6000 possessed advanced sequencing technology and was used for transcriptome sequencing in this study after library inspection was qualified. The raw sequence data are available in NCBI-SRA (https://www.ncbi.nlm.nih.gov/sra (accessed on 23 August 2023)) under the accession number PRJNA947359.4.9. Validation was conducted using real-time quantitative PCR (RT-qPCR).

### 4.9. Statistical Analysis

Regression analysis was performed using SPSS V25, which is suitable for analysis of variance between groups. The data shown in the article and every treatment group were subjected to the experiment 3 times.

### 4.10. Primers

All the primers used in this study are listed in Appendix A.

## 5. Conclusions

In this work, we identified the Cd-induced gene *HIPP26* from *A. paniculata* and transformed it into *A. thaliana*. *ApHIPP26* overexpression plants displayed Cd tolerance and accumulation in physiological and biochemical performances through enhancing the activity of antioxidant enzymes and modifying the expression of genes involved in the plant hormone pathways. These findings will help reveal the mechanism of Cd tolerance in hyperaccumulators, which may contribute to phytoremediation and even provide useful candidate molecular markers for breeding low-cadmium crops. As we all know, heavy metals have a strong toxic effect on organisms, seriously threatening our health, so reducing the pollution and accumulation of heavy metals is extremely important. In future studies, we may be able to find more hyperaccumulators to achieve the remediation of soil and water. It is also possible to transfer the key genes in the hyperaccumulators or wild relatives of cultivated species [75,76] into crops through transgenic technology to cultivate crops that are resistant to heavy metals and can be enriched in non-edible parts.

## Figures and Tables

**Figure 1 ijms-24-15052-f001:**
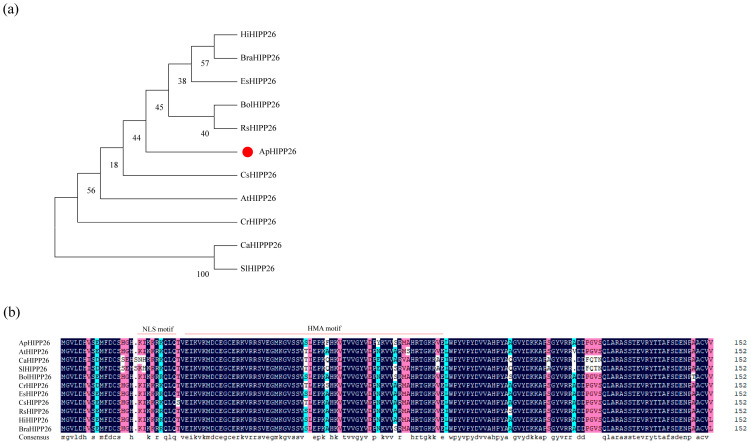
Phylogenetic relationship (**a**) and multiple sequence alignment (**b**) between ApHIPP26 and other plant homologous proteins.

**Figure 2 ijms-24-15052-f002:**
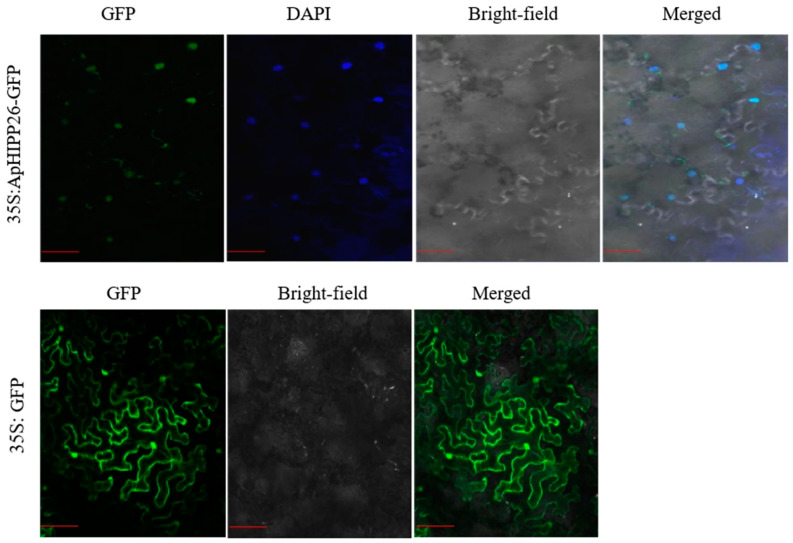
ApHIPP26 is localized in the nucleus. The 35S:*ApHIPP26*-GFP fusion protein was transiently expressed in tobacco. Scale bar = 50 μm.

**Figure 3 ijms-24-15052-f003:**
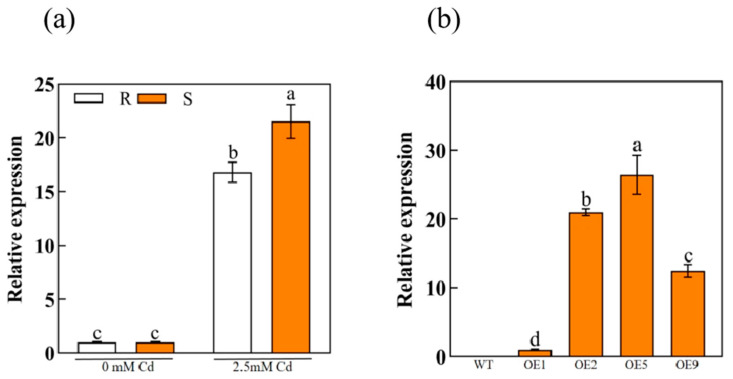
Expression patterns of the *ApHIPP26* gene. (**a**) Relative expression of *ApHIPP26* under cadmium (Cd) stress in *Arabis paniculata*. (**b**) Relative *ApHIPP26* expression in transgenic *Arabidopsis* overexpressing the *ApHIPP26* gene. Different letters indicate a statistically significant difference at *p* < 0.01.

**Figure 4 ijms-24-15052-f004:**
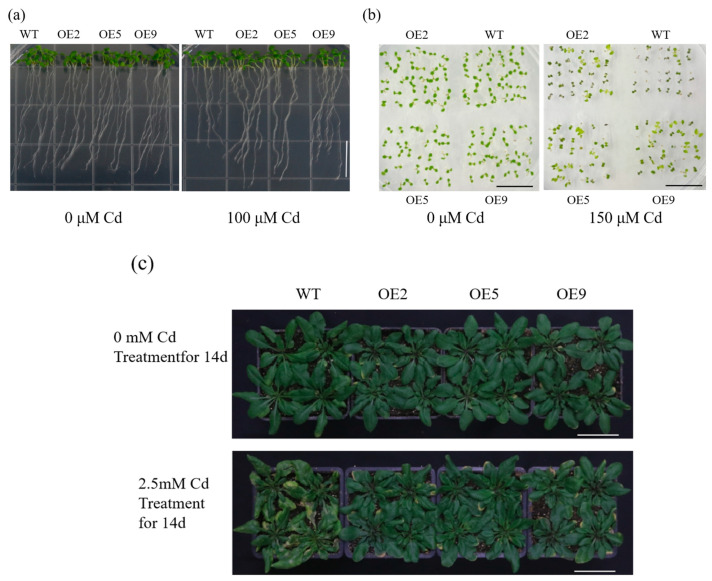
The growth state of *Arabidopsis* overexpressing the *ApHIPP26* gene under cadmium (Cd) stress. (**a**,**b**) Root length and Germination rate of transgenic *Arabidopsis* grown for 14 days on half-strength Murashige and Skoog (1/2 MS) medium without or containing 100 µM and 150 µM Cd. Scale bar = 1.5 cm. (**c**) Phenotypes morphology of wild-type and transgenic *Arabidopsis* treated with 0 or 2.5 mM Cd for 14 days. Scale bar = 5 cm.

**Figure 5 ijms-24-15052-f005:**
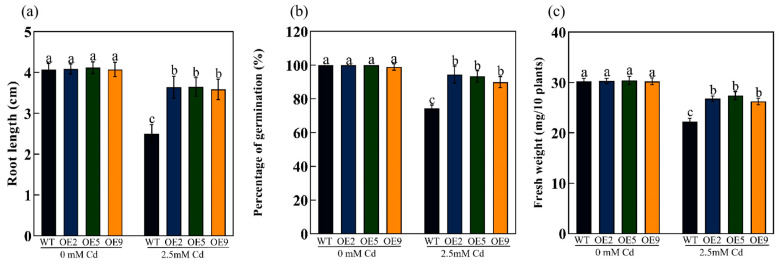
Detection of root length, percentage of germination, and fresh weight of WT and transgenic Arabidopsis overexpressing the *ApHIPP26* gene under cadmium (Cd) stress. (**a**) Root length. (**b**) percentage of germination. (**c**) Fresh weight. Each line after 14 days of growth in media with or without 100 μM Cd. Different letters indicate a statistically significant difference at *p* < 0.01.

**Figure 6 ijms-24-15052-f006:**
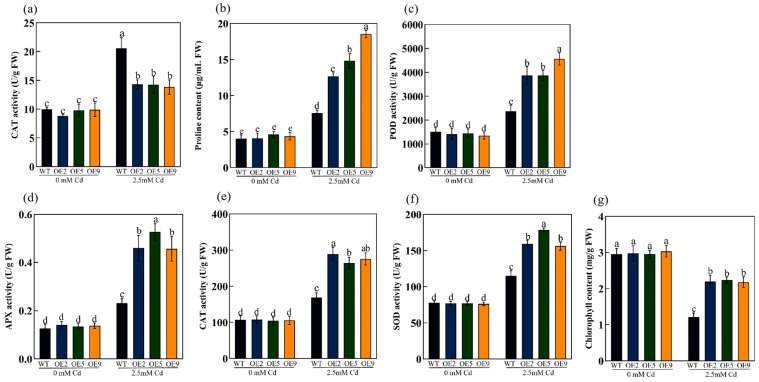
Physiological indices of plants expressing *ApHIPP26* under control and cadmium (Cd) stress conditions. (**a**) Malondialdehyde (MDA). (**b**) Proline (Pro). (**c**) Peroxidase (POD) activity. (**d**) Ascorbate peroxidase (APX) activity. (**e**) Catalase (CAT). (**f**) Superoxide dismutase (SOD). (**g**) Chlorophyll content. Different letters indicate a statistically significant difference at *p* < 0.01.

**Figure 7 ijms-24-15052-f007:**
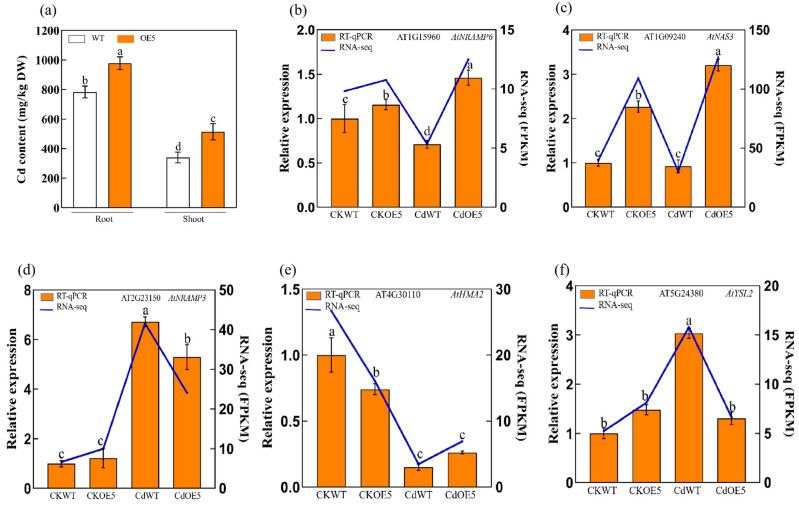
(**a**) Cadmium content in roots and buds of wild-type (WT) and transgenic *Arabidopsis* after 14 days of 2.5 mM Cd treatment. Relative expression levels of (**b**–**i**) heavy metal transporters in WT and transgenic lines under 0 mM or 2.5 mM Cd stress. Different letters indicate a statistically significant difference at *p* < 0.01.

**Figure 8 ijms-24-15052-f008:**
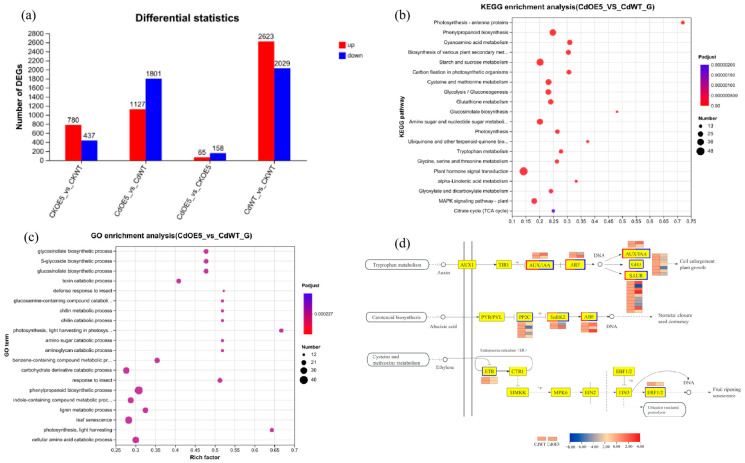
Differentially expressed genes (DEGs) in the shoots of wild-type (WT) plants and the transgenic line OE5 overexpressing *ApHIPP26* under cadmium (Cd) stress identified by transcriptome analysis. (**a**) The number of DEGs. (**b**) DEGs enriched in Kyoto Encyclopedia of Genes and Genomes (KEGG) pathways in the shoots of WT and OE5 plants affected by Cd stress. (**c**) Gene Ontology (GO) assignment of DEGs in the shoots of WT and OE5 plants affected by Cd stress. (**d**) Expression of DEGs related to Plant hormone signal transduction pathway.

**Figure 9 ijms-24-15052-f009:**
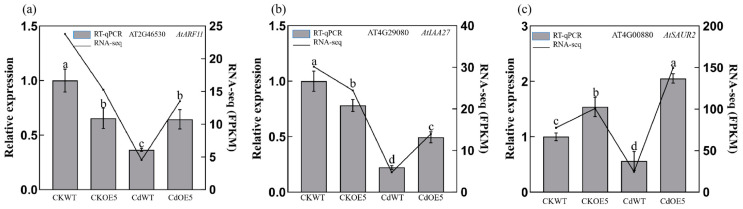
(**a**–**h**) RNA-seq gene validation was performed using qPC. According to the LSD test, the average FPKM value and relative expression level were shown. Different letters indicate a statistically significant difference at *p* < 0.01.

## Data Availability

Not applicable.

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
