# Peer review of "Overexpression of ApHIPP26 from the Hyperaccumulator Arabis paniculata Confers Enhanced Cadmium Tolerance and Accumulation to Arabidopsis thaliana"

_ijms, 2023, doi:10.3390/ijms242015052_

Round 1

Reviewer 1 Report

The manuscript by Zhou et al., titled “Overexpression of ApHIPP26 from the Hyperaccumulator Arabis paniculata Confers Enhanced Cadmium Tolerance and Accumulation to Arabidopsis thaliana” was focusing on the role of HIPP gene family; ApHIPP26 plays an important role in cadmium tolerance by alleviating oxidative stress and regulating plant hormones. This study is interesting and will be of interest to the readership of the IJMS journal. However, the followings are the specific comments:

Abstract:

Check the genes names “should be italic” in all manuscript.

Line 27, Arabidopsis, “changed to italic”.

Introduction:

Line 47, Therefore, it is urgent to study the tolerance mechanism of crops to cadmium – re-write this sentence.

Line 67-68, The function of HIPPs is apparent but not well studied, having been studied in only a handful of plants- check the grammar and re-write.

Results:

Line 93. Check all the scientific names, “should be italic” i.e., C. sativa.

Line 109-110, line 136; line 186; check the genes and scientific names.

Discussion, materials and methods

Please check the grammar and references.

The manuscript is commendably well-written and effectively presented. Each section demonstrates excellence and offers valuable insights for future research endeavors. These revisions, once addressed, will enhance the overall clarity and precision of the manuscript. Sincerely appreciate the authors' diligent efforts and dedication to this work, with these minor adjustments, the manuscript will be ready for publication in our journal.

Minor editing of English language required

Author Response

Dear Editor and Reviewers,

Thank you very much for your E-mail of “2-Oct-2023” regarding our manuscript ID “ijms-2632121”. Sincere thanks also go to the responsible and kind reviewers for helping us improve our manuscript in both scientific and linguistic aspects. As soon as receiving your E-mail, all the authors discussed the reviewers’ comments one by one carefully. We totally agree with the reviewers that major revisions need to make in our manuscript. Based on the instructions provided in your letter, we have made corrections which we hope meet with approval and  uploaded the file of the revised manuscript. The main corrections in the paper and the responds to reviewers’ comments are as following.

We would like to thank you for allowing us to resubmit a revised copy of the manuscript.

Sincerely,

Hongcheng Wang

Reviewer #1:

Comment 1: Check the genes names “should be italic” in all manuscript.

Response 1: Thank you very much for your careful correction. We carefully examined this manuscript and modified the genes names and wrote in italics.

Comment 2: Line 27, Arabidopsis, “changed to italic”.

Response 2: Thank you for your careful correction. We have revised it in this manuscript.

Comment 3: Line 47, Therefore, it is urgent to study the tolerance mechanism of crops to cadmium – re-write this sentence.

Response 3: We are grateful for the suggestion. “Therefore, it has been an urgent and high-priority issue to elucidate the molecular mechanism of Cd tolerance in plants and to breed or engineer crops with higher tolerance but less accumulation of Cd”. This is the sentence we rewrite, which is also reflected in the manuscript.

Comment 4: Line 67-68, The function of HIPPs is apparent but not well studied, having been studied in only a handful of plants- check the grammar and re-write.

Response 4: We appreciated the reviewer’s attention to detail. “The function of HIPPs is apparent but only characterized in a handful of plants”. This is the sentence we rewrite, which is also reflected in the manuscript.

Comment 5: Line 93. Check all the scientific names, “should be italic” i.e., C. sativa.

Response 5: Thank you for your careful correction. We have revised it in this manuscript.

Comment 6: Line 109-110, line 136; line 186; check the genes and scientific names.

Response 6: Thank you very much for your careful correction. We have modified your suggestion.

Comment 7: Please check the grammar and references.

Response 7: We are grateful for the suggestion. We examined the grammar and references in the manuscript very carefully and revised them.

Comment 8: Minor editing of English language required

Response 8: We appreciated the reviewer’s suggestion. The paper was polished by native English speaking professionals.

Reviewer 2 Report

propose an article entitled “Overexpression of ApHIPP26 from the Hyperaccumulator Arabis paniculata Confers Enhanced Cadmium Tolerance and Accumulation to Arabidopsis thaliana”. The manuscript is original in the conceptual idea, well structured, with new data relating on Cadmium (Cd) that is a wellknown toxic heavy metal that after accumulation in plants, and it also very dangerous on humans. The study shows as the Cd-induced gene ApHIPP26, located in the nucleus, from the hyperaccumulator Arabis paniculata and that compared with wild-type lines, Cd accumulated much more in transgenic Arabidopsis both aboveground and underground. Also under Cd stress, the activities of antioxidant enzymes  in the transgenic lines were higher than those in the wild type. The physiological and biochemical indices showed that the proline and chlorophyll contents in the transgenic lines increased significantly after Cd treatment, while the malondialdehyde (MDA) content decreased. The reserach shows that ApHIPP26 plays an important role in cadmium tolerance by alleviating oxidative stress and regulating plant hormones, which provides a basis for understanding the molecular mechanism of cadmium tolerance in plants and provides new insights for phytoremediation in Cd-contaminated areas.

1. Introduction

·         Lines 36-37. Heavy metals (HMs) are one of the most critical abiotic factors affecting soil and water and have become a serious worldwide problem [choose reference];

·         Lines 40-41. Among all kinds of heavy metal pollution, cadmium (Cd) is one of the most harmful [choose reference];

·         Line 49. Remember that Fe and Zn are microelement huseful for human healph while cd not; Please spend two words for this concept; for example in the genus Aegilops, that are Crop Wild Relatives of cultivated wheat (e.g. Perrino et al. 2014)

·         Line 69. Please add the author name when cite for the first time the scientific plant name. Check whole document in this way!

Ø  Camellia sinensis (L.) Kuntze

Ø  Oryza sativa….

Ø  Fagopyrum tataricum…..

Reference to be added:

ü  Perrino, E.V.; Brunetti, G.; Farrag, K. Plant communities of multi-metal contaminated soils: a case study in National Park of Alta Murgia (Apulia Region - southern Italy). International Journal of Phytoremediation 2014, 16, 871-888. Doi: 10.1080/15226514.2013.798626

ü  Perrino, E.V.; Wagensommer, R.P.; Medagli, P. The genus Aegilops L. (Poaceae) in Italy: Taxonomy, geographical distribution, ecology, vulnerability and conservation. Syst. Biodivers. 2014, 12, 331–349, Doi: 10.1080/14772000.2014.909543.

2. Results

Well done. Only few observation

·         Figures 1, 5 and 8. Please increase the font size, now is unclear.

3. Discussion

·         Lines 228-229. Recently, the study of hyperaccumulators has shed new light on these issues of heavy metal pollution and has become one of the research highlights. A recent study has demonstrated the existence of a relationship between the type of vegetation and the type and quantity of heavy metals present in the soil, which could lend a great hand for future phytoremediation operations [Perrino et al. 2014]

4. Materials and Methods

·         Figure 1. Well done but please specify the locxation of you detected A. thaliana, A. paniculata and Nicotiana tabacum …. Better if with geographical map specifing the system used (e.g. WGS84….) of Latitude and Longitude;

5. Conclusion

The conclusions are good, but I would conclude with the prospective research in this field

Reference

Please add Doi in each reference if  possible and available.

Author Response

Dear Editor and Reviewers,

Thank you very much for your E-mail of “2-Oct-2023” regarding our manuscript ID “ijms-2632121”. Sincere thanks also go to the responsible and kind reviewers for helping us improve our manuscript in both scientific and linguistic aspects. As soon as receiving your E-mail, all the authors discussed the reviewers’ comments one by one carefully. We totally agree with the reviewers that major revisions need to make in our manuscript. Based on the instructions provided in your letter, we have made corrections which we hope meet with approval and  uploaded the file of the revised manuscript. The main corrections in the paper and the responds to reviewers’ comments are as following.

We would like to thank you for allowing us to resubmit a revised copy of the manuscript.

Sincerely,

Hongcheng Wang

Responds to reviewers’ comments

Reviewer #2:

Comment 1: Lines 36-37. Heavy metals (HMs) are one of the most critical abiotic factors affecting soil and water and have become a serious worldwide problem [choose reference].

Response 1: We are grateful for the suggestion.  We added the following references:

Khoudi, H. Significance of Vacuolar Proton Pumps and Metal/H+ Antiporters in Plant Heavy Metal Tolerance. Physiologia Plantarum 2021, ppl.13447, doi:10.1111/ppl.13447.

Comment 2: Lines 40-41. Among all kinds of heavy metal pollution, cadmium (Cd) is one of the most harmful [choose reference]

Response 2: We are very grateful to you for pointing out our shortcomings. We realized that such a statement is not appropriate, and we modified and added references in the manuscript.

Akindele, E.O.; Omisakin, O.D.; Oni, O.A.; Aliu, O.O.; Omoniyi, G.E.; Akinpelu, O.T. Heavy metal toxicity in the water col-umn and benthic sediments of a degraded  tropical stream. Ecotoxicology and Environmental Safety. 2020, 190, 110153, doi:10.1016/j.ecoenv.2019.110153

Comment 3: Line 49. Remember that Fe and Zn are microelement useful for human health while Cd not; Please spend two words for this concept; for example in the genus Aegilops, that are Crop Wild Relatives of cultivated wheat (e.g. Perrino et al. 2014)

Response 3: We appreciated the reviewer’s attention to detail. Cd, Ca, Zn, and Fe are all bivalent ions with similar structures. The transporter transports heavy metal Cd into the body while transporting trace elements, but Cd is toxic to plants. Our expression is not clear, which has been modified in the manuscript. At the same time, we read the papers provided by reviewer, and these papers helped to improve our manuscript, we added references in the section of conclusion.

Comment 4: Line 69. Please add the author name when cite for the first time the scientific plant name. Check whole document in this way!

Response 4: We thank you for raising this useful point. We have modified the manuscript according to this format.

Comment 5: Figures 1, 5 and 8. Please increase the font size, now is unclear.

Response 5: We thank the reviewer for this insightful comment. We have increased the font size in figures.

Comment 6: Lines 228-229. Recently, the study of hyperaccumulators has shed new light on these issues of heavy metal pollution and has become one of the research highlights. A recent study has demonstrated the existence of a relationship between the type of vegetation and the type and quantity of heavy metals present in the soil, which could lend a great hand for future phytoremediation operations [Perrino et al. 2014]

Response 6: We appreciated the reviewer’s attention to detail. We have modified this sentence and added references.

Comment 7:  Figure 1. Well done but please specify the locxation of you detected A. thaliana, A. paniculata and Nicotiana tabacum …. Better if with geographical map specifing the system used (e.g. WGS84….) of Latitude and Longitude;

Response 7: We appreciated the reviewer’s attention to detail. A.thaliana, A.paniculata and Nicotiana tabacum used in this experiment were preserved in our laboratory(Hua xi District, China, E106.67°, N26.41°).

Comment 8: The conclusions are good, but I would conclude with the prospective research in this field.

Response 8: Thank you for your question. As we all know, heavy metals have a strong toxic effect on organisms, threatening our health, reducing heavy metal pollution and its importance. In future studies, we may be able to find more hyperaccumulators to achieve the remediation of soil and water. It is also possible to transfer the key genes in the hyperaccumulator or crop wild relatives of cultivated species into crops through transgenic technology to cultivate crops that are resistant to heavy metals and can be enriched in non-edible parts.

Reference added:

ü  Perrino, E.V.; Brunetti, G.; Farrag, K. Plant communities of multi-metal contaminated soils: a case study in National Park of Alta Murgia (Apulia Region - southern Italy). International Journal of Phytoremediation 2014, 16, 871-888. Doi: 10.1080/15226514.2013.798626

ü  Perrino, E.V.; Wagensommer, R.P.; Medagli, P. The genus Aegilops L. (Poaceae) in Italy: Taxonomy, geographical distribution, ecology, vulnerability and conservation. Syst. Biodivers. 2014, 12, 331–349, Doi: 10.1080/14772000.2014.909543.

Comment 8: Please add Doi in each reference if  possible and available.

Response 8: I 'm very sorry, because this journal only has ISSN number, and no separate doi number, so we can provide only ISSN(2349-2805) number.

Reviewer 3 Report

I described all the comments and note to attached PDF file, Please check the PDF file carefully. Figure 5~9, should be improved for the reader. 

Author Response

Dear Editor and Reviewers,

Thank you very much for your E-mail of “2-Oct-2023” regarding our manuscript ID “ijms-2632121”. Sincere thanks also go to the responsible and kind reviewers for helping us improve our manuscript in both scientific and linguistic aspects. As soon as receiving your E-mail, all the authors discussed the reviewers’ comments one by one carefully. We totally agree with the reviewers that major revisions need to make in our manuscript. Based on the instructions provided in your letter, we have made corrections which we hope meet with approval and  uploaded the file of the revised manuscript. The main corrections in the paper and the responds to reviewers’ comments are as following.

We would like to thank you for allowing us to resubmit a revised copy of the manuscript.

Sincerely,

Hongcheng Wang

Responds to reviewers’ comment

Reviewer #3:

Comment 1: Is it available for compare the arabidopsis HIPP26? And HIPP family protein itself has NLS(Nuclear Localization Signal) motif, so it would be better indicate in this figure NLS. One more isoprenylation site also.

Response 1: We thank the reviewer for this insightful comment. In the figure 1, we added the protein sequence of AtHIPP26 and identified the NLS motif.

Comment 2: As, I mentioned in the Fig. 1B legend, HIPP family has NLS, it can be expected the location of HIPP26 will be in nucleus.

Response 2: We thank you for raising this useful point. This is also consistent with our result.

Comment 3: WT doen't have any expression? in the Figure 3 (a), the authors shows the expression of WT expression.

Response 3: We thank you for raising this useful point. In Figure 3 (a), we performed the expression pattern of ApHIPP26 in roots and shoots of Arabis paniculata, and Figure 3 (b) is the expression pattern of ApHIPP26 after transformation into Arabidopsis thaliana. We detected the expression of HIPP26 from Arabis paniculata, there was no expression of ApHIPP26 in WT, but it was expressed in OE lines.

Comment 4: In the Materials and Methods, the concentration of Cd is 0.25mM. Which one is correct?

Response 4: We appreciated the reviewer’s attention to detail. The concentration used for Arabidopsis was 2.5 mM and Arabidopsis was cultured in pot, and the concentration for Arabis paniculata was 0.25 mM and it was cultured with hydroponics.

Comment 5: a and b, respectively, shows left one and right one, which one is 1/2MS or Cd treated figure?

Response 5: We are grateful for the suggestion. Both a and b, the left image is 1/2MS, and the right image is Cd treatment. We have marked in the manuscript.

Comment 6: 100uM for germination check? and 150uM for root length check? or reverse condition?? Why authors use different concentration?

Response 6: We appreciated the reviewer’s attention to detail. 100μM for root length check, and 150μM for germination check. Since Arabidopsis can only germinate at 150μM, the roots cannot grow normally. So we chose 100μM for the root length check.

Comment 7: It's not a stem morphology, rosette leaf phenoty or even simply 'phenotype' will be looks better.

Response 7: We sincerely thank you for your suggestion. We made careful modifications in the manuscript.

Comment 8: In figure 4 (a), seedling phenotype is 14 day, but same 14 day for leaf phenotype? Please check the periods for growth correctly.

Response 8: We are grateful for the suggestion. In Fig.4a, we placed Arabidopsis seeds after 1/2MS, vernalized for 3 days, and grew normally for 11 days. In Fig.4c, we transferred Arabidopsis seedlings to nutrient soil and cultured them normally for 21 days, and the phenotype after 14 days of Cd treatment. And we added the description in section of materials and methods.

Comment 9: treated how long days?

Response 9: Thank you for your careful correction. Treated for 14 days, and we added the description in Figure.

Comment 10: In the figure 4 a, b, authors use 100uM or 150uM Cd concentration. But in here, why use 1,000 higher concentration? From the authors data, seedling can not be germinate and grow for root length measurements.

Response 10: We are very grateful to you for pointing out our shortcomings. I am very sorry for our carelessness, as shown in Figure 4a, b, we used 100um Cd. We've made serious changes in the manuscript.

Comment 11: Check please, all the concentration mentioned above is 2.5mM.

Response 11: We appreciated the reviewer’s attention to detail. We expanded the description in section of materials and methods and confirmed the concentrations emerged in different sections.

Comment 12: Figure 5~9, should be improved for the reader.

Response 12: We thank you for raising this useful point. We have revised it carefully according to your opinion.

Thank you again for your positive and constructive comments and suggestions on our manuscript.

Round 2

Reviewer 2 Report

Dear authors,

I appreciate the work done in this last version of the manuscript